

# Are new open building data useful for flood vulnerability modelling?

Marco Cerri[1], Max Steinhausen[1], Heidi Kreibich[1], and Kai Schröter[1]

[1]German Research Centre for Geosciences GFZ, Section Hydrology, Telegrafenberg, 14473 Potsdam, Germany

**Correspondence:** Kai Schröter (kai.schroeter@gfz-potsdam.de)

**Abstract.**

Flood risk modelling aims to quantify the probability of flooding and the resulting consequences for exposed elements. The assessment of flood damage is a core task which requires the description of complex flood damage processes including the influences of flooding intensity and vulnerability characteristics. Multi-variable modelling approaches are better suited for this purpose than simple stage-damage functions. However, multi-variable flood vulnerability models also often have problems to predict damage for regions other than those for which they have been developed. A transfer of vulnerability models usually results in a drop of model predictive performance. Here we investigate the question whether data from the open data source OpenStreetMap is suitable to model flood vulnerability of residential buildings and whether the underlying standardized data model is helpful to transfer models across regions. We develop a new data set by calculating numerical spatial measures for residential building footprint geometries and combine these variables with an empirical data set of observed flood damage. From this data set random forest regression models are learned using regional sub-sets and are tested for predicting flood damage in other regions. This regional split-sample validation approach reveals that the predictive performance of models based on OpenStreetMap data is comparable to alternative multi-variable models, which use comprehensive and detailed information about preparedness, socio-economic status and other aspects of residential building vulnerability. However, our results show that using numerical spatial measures derived from OpenStreetMap building geometries does not resolve all problems of model transfer. Still, we conclude that these variables are useful proxies for flood vulnerability modelling, because these data are consistent, openly accessible, and thus make it easier and more cost-effective to transfer vulnerability models to other regions.

## 1 Introduction

Floods have huge socio-economic impacts globally. Driven by increasing exposure, as well as increasing frequency and intensity of extreme weather events, consequences of flooding have sharply risen during recent decades (Hoeppe, 2016; Lugeri et al., 2010). Therefore, effective adaptation to growing flood risk is an urgent societal challenge (UNISDR, 2015; Jongman, 2018). With the transition to risk oriented approaches in flood management, flood risk models are important tools to conduct





quantitative risk assessments as a support for decision making from continental to local scales (Alfieri et al., 2016; Moel et al., 2015; Winsemius et al., 2013). While macro- or meso-scale risk assessment approaches target regional, national or continental studies, risk assessment on the micro-scale is needed to guide urban planning, optimize investment for protection and other mitigation measures considered in flood risk management plans (Meyer et al., 2013; Moel et al., 2015; Rehan, 2018). Flood risk

models include components to represent the key elements of flood risk: hazard, exposure and vulnerability (Kron, 2005). Flood hazard is usually modeled with high spatial resolutions in order to realistically capture variability in flood hazard intensity in consideration of local topographic characteristics (Apel et al., 2009; Teng, 2017). For consistent risk assessments, exposure and vulnerability need to be analysed on similar scales and with appropriate spatial resolution. With an increasing availability of new exposure data sets including for instance information about the number, occupancy, and characteristics of exposed objects

(Figueiredo and Martina, 2016; Paprotny et al., 2020; Pittore et al., 2017) micro-scale exposure and vulnerability modeling gains much traction (Schröter et al., 2018; Lüdtke et al., 2019; Sieg et al., 2019).

    Both synthetic (e.g. Blanco-Vogt and Schanze (2014); Dottori et al. (2016); Penning-Rowsell and Chatterton (1977)) and empirically based models (e.g. Thieken et al. (2005); Zhai et al. (2005)) have been proposed for micro-scale vulnerability modelling. As flood damaging processes are complex, a large diversity of influencing factors needs to be taken into account to

capture and appropriately represent flooding intensity and resistance characteristics of exposed elements in flood vulnerability models (Thieken et al., 2005). In this context, multi-variable modelling approaches are an important advancement from simple stage-damage curves, which relate only inundation depth to flood loss. While multi-variable vulnerability models usually outperform traditional stage-damage functions (Merz et al., 2004; Schröter et al., 2014), the downside of these approaches is an increased need of detailed data on the level of individual objects (Merz et al., 2010, 2013) which are often not available in

the target area of the analysis (Apel et al., 2009; Cammerer et al., 2013; Dottori et al., 2016). Missing standards for collecting comparable and consistent data are one reason for this problem (Changnon, 2003; Meyer et al., 2013). Hence, providing the input variables for multi-variable flood vulnerability models on the micro-scale is a key challenge for their practical applicability. Another challenge is the generalization of locally derived vulnerability models. A number of studies confirm a model performance mismatch between regions where models have been developed and the target areas for application (Cammerer

et al., 2013; Jongman et al., 2012; Schröter et al., 2016; Wagenaar et al., 2018). It is argued that the generalized application of vulnerability models to different geographic and socio-economic conditions needs to consider an adequate representation of local characteristics and damage processes (Felder et al., 2018; Figueiredo et al., 2018; Sairam et al., 2019). Hence, consistency in input data is an important requirement for the spatial transfer of vulnerability models (Lüdtke et al., 2019; Molinari et al., 2020). The availability, accessibility and consistency of data sources are important requirements for generalized vulnerability

model applications but also poses requirements on modelling approaches. With an increased number of input variables and an enlarged diversity of data sources used for vulnerability modelling, we usually deal with heterogeneous data in terms of different scaling, degrees of detail, resolution and complex inter-dependencies (Schröter et al., 2016, 2018). Tree based algorithms are a suitable approach to handle heterogeneous data, represent non-linear and non-monotonic dependencies, and, as a nonparametric approach, do not require assumptions about independence of data (Carisi et al., 2018; Merz et al., 2013; Schröter

et al., 2014; Wagenaar et al., 2017). The Random Forest (RF) algorithm (Breiman, 2001) is broadly used in many disciplines,



due to its high predictive accuracy, simplicity in use and flexibility concerning input data. In the domain of flood risk modelling, Wang et al. (2015) have successfully applied RF for flood risk assessment and Bui et al. (2020) used RF for flood susceptibility mapping. Merz et al. (2013) demonstrated the suitability of tree based algorithms for flood vulnerability modelling. Following this, Carisi et al. (2018); Chinh et al. (2015); Hasanzadeh Nafari et al. (2016); Sieg et al. (2017); Wagenaar et al. (2017) have

used RF and other tree based algorithms for flood loss estimation in flood prone regions in Vietnam, Australia, the Netherlands and Italy. In these studies, vulnerability modelling using RF was based on site specific empirical data sets which had been collected ex-post major flood events. In contrast, the framework proposed by (Amirebrahimi et al., 2016) successfully used 3D building information for flood damage assessment of individual buildings. Gerl et al. (2016) and Schröter et al. (2018) investigated the suitability of alternative more general data sources for flood vulnerability modelling using urban structure type

information derived from remote sensing images, virtual 3D city models and numerical spatial measures which describe the extent and shape complexity of residential buildings. It was shown that particularly geometric information about buildings as for instance building area and height are useful variables to describe building characteristics relevant for estimating flood losses (Schröter et al., 2018). From these studies it has been concluded that data about building footprint geometry work as a proxy to describe resistance characteristics of buildings. However, further analyses are needed to understand whether building geometry

data enable consistent flood vulnerability modelling with high resolution and are suitable to characterise differences in flood vulnerability across regions. With new data sources emerging from crowdsourcing projects and open data initiatives, detailed building data are increasingly available and accessible (Irwin, 2018). Open and/or standard building data are a promising data source to coherently describe exposure and characterise vulnerability of residential buildings, and to improve the spatial transfer of vulnerability models given a consistent underlying data model and clear specification of input variables across regions.

Data science methods are predestined to make use of these data in flood vulnerability modelling. Against this backdrop, we investigate the suitability of the open data source OpenStreetMap (OSM) (contributors, 2020) for flood vulnerability modelling of residential buildings. OSM is a geographic database with a worldwide coverage which is nowadays considered as reliable for most of civil and common usages (Barrington-Leigh and Millard-Ball, 2017). The information about building footprints is freely available and straightforward to obtain from public online servers. The OSM contributors' community is constantly

growing and assures regular updates in terms of accuracy and completeness of the data (Hecht et al., 2013).

We test the hypothesis that numerical spatial measures derived from OSM building footprint geometries provide useful information for the estimation of flood losses to residential buildings. From the underlying consistent OSM data model and standardized calculation of spatial measures we expect an improvement of the spatial transfer of flood vulnerability models across regions. Accordingly, the research objectives are i) to understand which building related variables are useful to describe

building vulnerability, ii) to learn predictive flood vulnerability models, and iii) to test and evaluate model transfer across regions. In Section 2 the data sources, the derived variables and the preparation of data sets are described. Section 3 introduces the methods to identify predictor variables and to derive predictive modelsi. Further, it describes the set-up for testing and evaluating model performance in spatial transfers. The results from this analysis are reported and discussed in Section 4. Conclusions are drawn in Section 5.


## 2  Data

We use an empirical data set of relative loss to residential buildings and influencing factors which has been collected via computer aided telephone interviews (CATI) during survey campaigns after major floods in Germany since 2002. Another data source is OSM (contributors, 2020) providing information about building locations, geometries, occupancy and other characteristics. OSM data is complemented with numerical spatial measures calculated from OSM building footprint geometries.

### 2.1  Computer aided telephone interview data

Computer aided telephone interview (CATI) surveys were conducted with affected private households ex-post major floods in Germany. The regional focal points of flood impacts were the Elbe catchment in east Germany, and the Danube catchment in southern Germany. Particularly noteworthy are the floods of 2002 and 2013, which caused economic losses of EUR 11.6 bn (reference year 2005) and EUR 8 bn respectively in Germany (Thieken et al., 2006, 2016). With EUR 1 bn economic damage, the city of Dresden at the Elbe River in Saxony has been a hotspot of flood impacts during the August 2002 flood (Kreibich and Thieken, 2009). In August 2002, flash floods triggered by record breaking precipitation and numerous levee failures caused widespread flooding along the Elbe River and its tributaries in Saxony and Saxony-Anhalt as well as along the Regen River and other southern tributaries to the Danube River in Bavaria (Schröter et al., 2015). The magnitude of flood peak discharges along these rivers well exceeded a statistical return period of 100 years (Ulbrich et al., 2003). In May 2013 a pronounced precipitation anomaly with subsequent extreme precipitation end of May/beginning of June caused severe flooding in June 2013 especially along the Elbe and Danube rivers with new water level records and major dike breaches both at the Elbe and Danube Rivers (Conradt et al., 2013; Merz et al., 2014; Schröter et al., 2015). The magnitude of flood peak discharges exceeded statistical return periods of 100 years along the Elbe, Mulde and Saale tributaries, and along the Danube and Inn River in Bavaria (Blöschl et al., 2013; Schröter et al., 2015). With 180 questions, the CATI surveys cover a broad range of flood impact related factors including building characteristics, effects of warnings, precaution and the socio-economic background of households. The survey campaigns for different floods are consistent in terms of acquisition methodology, type and scope of questions. The interviewees were randomly selected from lists of potentially affected households along inundated streets which have been identified from satellite data, flood reports and press releases. With an average response rate of 15%, in total 3056 interviews have been completed. For further details about the surveys and data processing refer to (Kienzler et al., 2015; Thieken et al., 2005, 2017). Building on the findings of previous work (Merz et al., 2013; Schröter et al., 2014), for this study 23 variables have been preselected with a focus on building characteristics, flood intensity at the building, socio-economic status as well as warning, precaution and previous flood experience (Table 1). In addition, relative loss to the building has been determined as the ratio of reported actual losses and the building value (replacement cost) at the time of the flood event (Elmer et al., 2010). Hence, it describes the degree of building damage on a scale from 0 (no damage) to 1 (total damage). Building values are based on the standard actuarial valuation method of the insurance industry in Germany (Dietz, 1999) which estimates replacement costs using information about the floor space, basement area, number of storeys, roof type, etc. that are available from CATI data. Relative loss to the building (*rloss*), and water depth (*wst*) at the building are the key variables from the CATI dataset


used in this study. *rloss* is used to learn predictive models and to evaluate their performance. Consequently, the records in the CATI data set without values for *rloss* are removed. This reduces the number of available records from 3056 to 2203. *wst* is the most commonly used predictor in flood vulnerability modelling (Gerl et al., 2016), because it is a highly relevant characteristic of flood intensity and it is usually available from hydrodynamic-numeric simulations. *wst* from CATI is a continuous variable

with a length unit in centimeters. Negative values represent a water level below the ground surface, which affects only the basement of a building.

## 2.2  OpenStreetMap data

OSM is a free web-based map service built on the activity of registered users who contribute to the database by adding, editing or deleting features based on their local knowledge. The contributors use GPS devices and satellite as well as aerial imagery

to verify the accuracy of the map. OSM is an open data project and the cartographic information can be downloaded, altered and redistributed under the Open Data Commons Open Database License (ODbL) (contributors, 2020). Among the so-called volunteered geographic information (VGI) projects (Goodchild, 2007), OSM is the most widely known. OSM data provide information about building locations, footprint geometries, occupancy and other characteristics. The positional accuracy of OSM data, and the completeness of the database in respect to the number of mapped objects present in the real world, are

nowadays considered as satisfactory for most of the developed countries and urban areas (Barrington-Leigh and Millard-Ball, 2017; Hecht et al., 2013). On the contrary, information on object attributes such as road names or building types are often scarce and inconsistent. The tag "building" is used to identify the outline of a building object in OSM. The majority of buildings (82%) has no further description and only 12% are specified as primarily "residential" or a single family "house" (https://taginfo.openstreetmap.org/keys/building#values (28.02.2020)). Therefore, the filtering for residential buildings from

the OSM database uses the underlying 'residential' landuse information of OSM. By joining the landuse information to the building polygons, those of residential occupation can be identified and selected.

## 2.3  Data preparation

The OSM and CATI data sets have been conflated in order to link the empirically observed variables *rloss* and *wst* with OSM data for individual residential buildings. This operation uses the geolocation information of both data sources. The CATI data

are provided with address details including community, zip code, street name, and the house number ranges in blocks of 5 numbers. Geocoding algorithms including open web API (Application Programming Interface) services like Google (developers.google.com/maps/documentation/geolocation), Photon (photon.komoot.de) and Nominatim (nominatim.openstreetmap.org) were applied to obtain geocoordinates for the address information from the interview data.

OSM is a spatial data set including georeferenced building outlines. The geolocated interviews are spatially matched with

OSM building polygons using an overlay operation which merges interview points with OSM building polygons. In view of limited address details regarding the building house number ranges and inherent inaccuracies of geocoding databases and algorithms (Teske, 2014) a buffer radius of 5 meters has been used to correct for offsets between geocoding points and building polygons. CATI records which still could not be matched with OSM geometries and with obviously erroneous geolocations,



**Table 1.** Preselected variables from CATI surveys; C: continuous, O: ordinal, N: nominal scaled variables

| | | Variable | Type and range |
|---|---|---|---|
| | | **Warning, precaution and previous experience** | |
| 1 | wt | Early warning lead time | C: 0 to 336 h |
| 2 | wq | Quality of warning | O: 1 = knew exactly what to do to 6=had no idea what to do |
| 3 | ws | Indicator of flood warning source | O: 0 = no warning to 4 = official warning through authorities |
| 4 | wi | Indicator of flood warning information | O: 0 = no helpful information to 11 = many helpful information |
| 5 | wte | Lead time period not used for emergency | C: 0 to 335 h |
| 6 | em | Emergency measures indicator | O: 1 = no measures undertaken to 17 = many measures undertaken |
| 7 | epre | Perception of efficiency of private precaution | O: 1 = very efficient to 6=not efficient at all |
| 8 | pre | Precautionary measures indicator | O: 0 = no measures undertaken to 38 = many, efficient measures undertaken |
| 9 | fe | Flood experience indicator | O: 0 = no experience to 9 = recent flood experience |
| 10 | kh | Knowledge of flood hazard | N (yes / no) |
| | | **Hydraulic characteristics of the inundation** | |
| 11 | wst | Water depth | C: 248 cm below ground to 670 cm above ground |
| | | **Building characteristics** | |
| 12 | bt | Building type | N (1 = multifamily house, 2 = semi-detached house, 3=one-family house) |
| 13 | nfb | Number of flats in building | C: 1 to 45 flats |
| 14 | fsb | Floor space of building | C: 45 to 18000 m$^2$ |
| 15 | bq | Building quality | O: 1=very good to 6 = very bad |
| 16 | bv | Building value | C: 92244 to 3718677 EUR |
| | | **Socio-economic status of the residents** | |
| 17 | age | Age of the interviewed person | C: 16 to 95 yrs |
| 18 | hs | Household size, i.e. number of persons | C: 1 to 20 people |
| 19 | chi | Number of children (<14 years) in household | C: 0 to 6 |
| 20 | eld | Number of elderly persons (>65 years) in household | C: 0 to 4 |
| 21 | own | Ownership structure | N (1 = tenant; 2 = owner of flat; 3 = owner of building) |
| 22 | inc | Monthly net income in classes | O: 11 = below 500 EUR to 16 = 3000 EUR and more |
| 23 | socP | Socio-economic status according to Plapp2003 | O: 3 = very low status to 13 = very high status |
| | | **Experienced damage** | |
| - | rloss | Relative loss of the residential building | C: 0 = no damage to 1 = total damage |



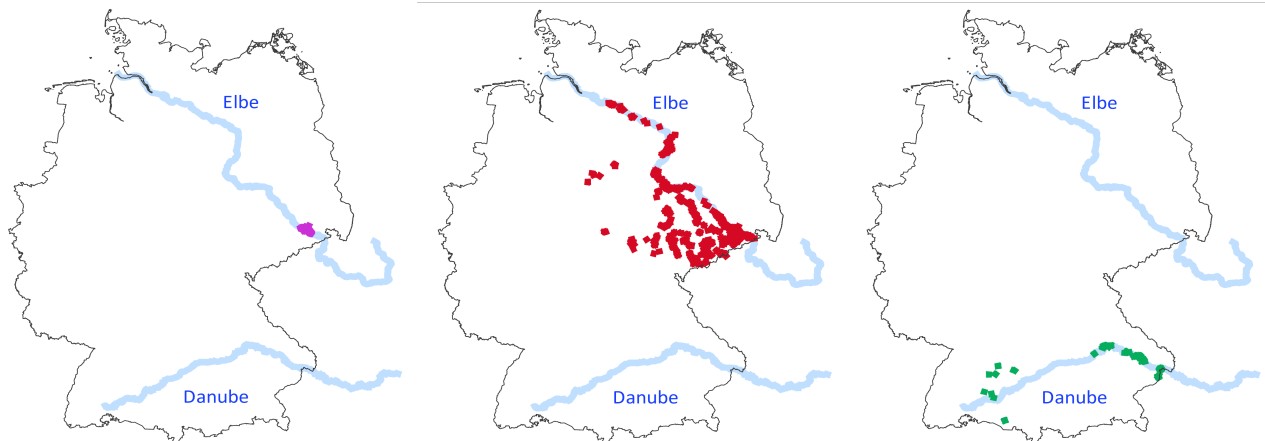

**Figure 1.** Regional sub division of dataset for spatial split sample testing (Dresden (left), the Elbe catchment (middle) and the Danube catchment (right))

e.g. position is far away from flood affected areas or urban settlements, have been removed from the data set. After these steps 1649 records remain from the original set of CATI surveys. The spatial distribution of these data points highly concentrates on the Elbe catchment (1234 records) including Dresden (310 records) and on the Danube catchment (105 records) (Fig. 1)

## 2.4 Numerical measures

Information about building geometry is useful to support the estimation of flood losses to residential buildings (Schröter et al., 2018). Building on this knowledge, numerical spatial measures are calculated for OSM building footprint geometries with the aim to add potential explanatory variables to the estimation of relative loss to residential buildings. For this purpose, image analysis algorithms typically used in landscape ecology are adopted. These algorithms calculate numerical spatial measures like area, perimeter, elongation and complexity based on the analysis of geometries identified in aerial or remote sensing images (Jung, 2016; Lang and Tiede, 2003; Rusnack, 2017). From the OSM building footprint geometries the following variables are determined: (1) area (Area) of the building polygon in square meters, (2) perimeter (Perimeter) in meter, (3) degree of compactness (DegrComp), level of compactness of the building polygon based on the relative distance of the internal vertex points, normalized to a circle and scaled from 0 to 1, (4) perimeter-area ratio (PARatio), basic measure of the complexity of the shape, but biased by the dimension of the polygon, (5) shape index (ShapeIndex), more accurate metric for the shape complexity because it does consider the polygon size, it is normalized to a square so its value is suitable for comparing buildings, (6) fractal dimension index (FracDimInd), alternative measure to evaluate the shape complexity of a polygon, considering the polygon size and normalized to a square, scaled from 1 to 2, (7) radius of gyration (RadGyras), measure to express the elongation of the polygon together with its dimension, in meter, (8) linear segment indicator (LinSegInd), ratio between the major and minor axis of the polygon, to give a measure of the shape elongation, normalized to a square, (9) ratio of bounding rectangle





area (BoundRatio), ratio between the area of the bounding rectangle and the area of the polygon, measure for the shape complexity, normalized to the corresponding bounding rectangle. The numerical spatial measures calculated for each OSM building polygon are compiled in Table 2 along with the other variables available from CATI that are used to derive flood vulnerability models. The meaning of these spatial measures, the equations as well as the range of values and examples are
listed in the Appendix A1.

**Table 2.** Variables of the amended OSM data set for each building object

| | Empirical variables from the CATI interviews | |
|---|---|---|
| - | Relative loss of the residential building (rloss) | Relative loss, 0 = no damage to 1 = total damage |
| - | Water depth (wst) | Water level respect to the ground level, - 248 cm to 670 cm |
| | Numerical spatial measures calculated for OSM building geometries | |
| 1 | Area (Area) | Area of the building, 0 m$^2$ to $\infty$ |
| 2 | Perimeter (Perimeter) | Perimeter of the building, 0 m to $\infty$ |
| 3 | Degree of compactness (DegrComp) | Compactness of the building shape, relative vicinity of the internal points, normalized to a circle, 0 to 1 |
| 4 | Perimeter-area ratio (PARatio) | Shape complexity, biased by building size, 0 to $\infty$ |
| 5 | Shape index (ShapeIndex) | Shape complexity, adjusted to building size, normalized to a square, 1 to $\infty$ |
| 6 | Fractal dimension index (FracDimInd) | Shape complexity, adjusted to building size, scaled between 1 and 2 |
| 7 | Radius of gyration (RadGyras) | Building extent and compactness, 0 m to $\infty$ |
| 8 | Linear segment indicator (LinSegInd) | Elongation of the polygon, normalized to a square, 1 to $\infty$ |
| 9 | Ratio of bounding rectangle area (BoundRatio) | Shape complexity, normalized to the hypothetical simplest polygon, 1 to $\infty$ |

## 3  Methods

We analyse the created data set with two main objectives. First, we strive to identify those variables from Table 2, which are most useful to explain relative loss to residential buildings. Second, we aim to derive flood vulnerability models for residential buildings and to test these models for spatial transfers across regions. The data analyses workflow including data
pre-processing, model learning, model selection and model transfer are illustrated in Fig. 2.

The data preparation and numerical spatial measures have been described in the previous section. For model learning and model transfer we use the Random Forest (RF) machine learning algorithm introduced by (Breiman, 2001). For variable selection and predictive model learning RF provide a concept to quantify the importance of candidate explanatory variables which allows to select the subset of most relevant variables. RF are also an efficient algorithm to learn predictive models from
heterogeneous datasets with complex interactions and with different scales like continuous or categorical information (Huang and Boutros, 2016). RF are an extension of the classification and regression tree (CART) algorithm (Breiman et al., 1984) which aims to identify a regression structure among the variables in the dataset. Regression trees recursively sub-divide the space of predictor variables to approximate a nonlinear regression structure. This sub-division is driven by optimizing the accuracy of





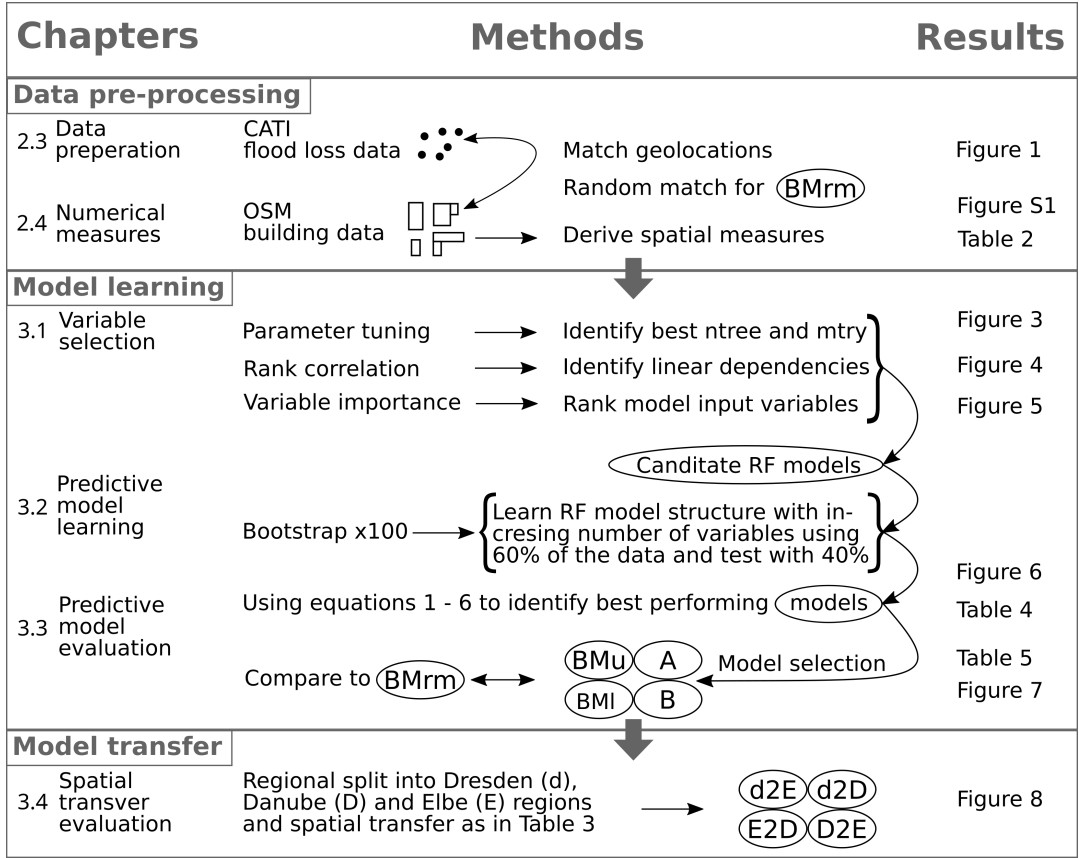

**Figure 2.** Data pre-processing, model learning and model transfer workflow

local regression in these regions which, by repeated partitioning, leads to a tree structure. Predictions are made by following the division criteria along the nodes and branches from the root node to the leaves which finally contain the predicted value for a given set of input variables. RF incorporates bootstrap aggregation (bagging) as a simple and powerful ensemble method to reduce the variance of the CART algorithm. In comparison to single trees, RF are more suitable to identify complex patterns and structures in the data (Basu et al., 2018). As an ensemble approach, RF learns a regression tree for a number of bootstrap

5 replica of the learning data. This results in a number of trees (*ntree*) forming a forest of regression trees. To reduce correlation between trees, the RF algorithm randomly selects a subset of variables (*mtry*) which are evaluated for dividing the space of predictor variables. This efficiently reduces overfitting and makes RF less sensitive to changes in the underlying data. Each bootstrap replica is created by randomly sampling with replacement about two thirds of observations from the original data set.

10 The remaining data are indicated as out-of-bag (OOB) observations and are used for evaluating the predictive accuracy of the tree, in terms of the OOB error. For regression trees the OOB error is the mean squared sum of residuals. For loss estimation,





the predictions of all trees are combined by aggregating the individual predictions as the mean prediction from the forest. The predictions of the individual trees, i.e. from the ensemble of models, provide an estimate of predictive uncertainty.

RF predictive model performance is sensitive to specifications of the algorithm parameters *mtry* and *ntree* (Huang and Boutros, 2016). Therefore, the optimum values for both parameters are identified as those which yield minimum OOB errors
on an independent data set. For parameter tuning, we pursue the variation approach implemented by (Schröter et al., 2018) by selecting parameters from a broad and comprehensive range of values $ntree \in [100, 500, 1000, 2000, 3000, \dots 15000]$ and $mtry \in [p/6, p/3, 2p/3]$ with p as number of candidate predictors and derive RF models for each combination. For each pair of chosen values, the algorithm is repeated 100 times to account for inherent data variability. The optimum parameters will minimize the prediction error on the OOB sample data. Using the optimum RF parameter settings, we derive predictive models
for *rloss*.

## 3.1 Variable selection

The first step in model learning is the selection of variables to be used as predictors in the model. The analysis of the Spearman's rank correlation between the variables gives a first insight into the linear dependency structure of the data-set. Furthermore, RF support the evaluation and ranking of potential predictors by quantification of variable importance which also accounts for
variable interaction effects. The importance of a selected variable is evaluated by calculating the changes of the squared error of the predictions when the values of that variable are randomly permuted in the OOB sample. The increase of the average error will be larger for more important variables and smaller for less important variables. On this basis it is possible to decide which variables to include in a predictive model. The outcomes of variable importance evaluations are sensitive to the RF algorithm parameters *mtry* and *ntree* (Genuer et al., 2010). Therefore, to achieve stable results for this analyses we implement a robust
approach which averages the outcomes of multiple runs with variations in RF parameters (Schröter et al., 2018): $ntree \in [500, 1000, 1500, 2000, \dots 5000]$ whereby each tree is repeatedly built for $mtry \in [p/6, p/3, 2p/3]$, with p as number of candidate predictors, which correspond to the lower limit, the default value and the upper limit, suggested by (Breiman, 2001). Following this procedure, the potential explanatory variables of our data set (Table 2) are evaluated and ranked according to their relative importance to predict *rloss*.

## 3.2 Predictive model learning

Variable selection needs to be considered as an essential part of the model selection process. Therefore, candidate RF models using different numbers of variables are assessed in terms of predictive performance for independent data.

The OSM based numerical spatial measures differentiate building form and shape complexity. To gain further insights into the suitability of these variables for flood vulnerability modelling we incrementally add explanatory variables to the learning
data set. Based on the outcomes of variable importance ranking the learning set is expanded variable by variable and models of increasing complexity are learned (c.f. Table 2. From the comparison of model predictive performance between these candidate models the best balance between model performance and number of input variables is assessed. This is implemented by bootstrapping splitting the data into sub-set for learning (60%) and testing (40%) with 100 iterations.



### 3.3 Predictive model evaluation

Model predictive performance is evaluated by comparing predicted ($P$) and observed ($O$) *rloss* values from the validation sample using the following metrics. In these metrics RF predictions are evaluated for the median prediction ($P_{50}$) derived from the ensemble of individual tree predictions.

Mean Absolute Error (MAE) quantifies the precision of model predictions, with smaller values indicating higher precision:

$$MAE = \frac{1}{n}\sum_{i=1}^{n}|P_{50_i} - O_i| \tag{1}$$

Mean Bias Error (MBE) is a measure of accuracy, i.e. systematic deviation from the observed value. Unbiased predictions yield a value of 0:

$$MBE = \frac{1}{n}\sum_{i=1}^{n}(P_{50_i} - O_i) \tag{2}$$

Mean Squared Error (MSE) combines the variance of the model predictions and their bias. Again, smaller values indicate better model performance:

$$MSE = \frac{1}{n}\sum_{i=1}^{n}(P_{50_i} - O_i)^2 \tag{3}$$

The ensemble of model predictions from the RF models offers insight into prediction uncertainty. This property is analyzed by evaluating the 90-percent quantile range, i.e. the difference between the 5-quantile and 95-quantile in relation to the median,

as a measure of ensemble spread:

$$QR_{90} = \frac{1}{n}\sum_{i=1}^{n}(P_{95_i} - P_{5_i})/P_{50_i} \tag{4}$$

with 95-quantile, 5-quantile and the 50-quantile, i.e. the median of the predictions. $QR_{90}$ is a measure of sharpness with smaller values indicating a smaller prediction uncertainty.

Reliability of model predictions is quantified in terms of the hit rate (Gneiting and Raftery, 2007):

$$HR = \frac{1}{n}\sum_{i=1}^{n}h_i \;\; ; \; h_i = \begin{cases} 1, \; if \; O_i \in [P_{95_i}, P_{5_i}] \\ 0, \; otherwise \end{cases} \tag{5}$$

$HR$ calculates the ratio of observations within the 95-5-quantile range of model predictions. For a reliable prediction $HR$ should correspond to the expected nominal coverage of 0.9.





$HR$ and $QR_{90}$ are combined to the interval score ($IS$) which accounts for the trade-off between $HR$ values and $QR_{90}$ ranges (Gneiting and Raftery, 2007):

$$IS = QR_{90} + \frac{1}{n}\sum_{i=1}^{n}\frac{2}{\beta}\left(P_{05_i} - O_i\right)|\{O_i < P_{05_i}\} + \frac{2}{\beta}\left(O_i - P_{95_i}\right)|\{O_i > P_{95_i}\} \qquad (6)$$

Further, an independent assessment of OSM based vulnerability model performance we consider two benchmark models.
We argue that the set of CATI variables (Table 1) represents the most detailed data set available for flood loss estimation of residential buildings (Merz et al., 2013; Schröter et al., 2014; Thieken et al., 2016). Therefore, a RF model is learned using all 23 CATI predictors as an upper benchmark (BMu). In contrast, a RF model using only water depth as a predictor is learned as a lower benchmark. The reasoning is, that using extra variables in addition to water depth will improve the predictive performance of the models (Schröter et al., 2018, 2016). As described in section 2.3, the detail of geolocation information from
CATI data is limited to ranges of house numbers. Therefore, we face uncertainty in whether CATI data and OSM building geometries have been matched correctly. To assess the potential implications of this source of uncertainty we derive a model (BMrm) which is based on a data set with *rloss* and *wst* observations randomly assigned to OSM building footprints. We keep the RF modelling approach for the benchmark models consistent to ensure that any observed difference in model performance stems from differences in the underlying input variables.

**3.4 Spatial transfer evaluation**

We investigate the question whether the consistent data basis of OSM based numerical spatial measures supports the transfer of flood vulnerability models across regions by splitting the available data set into subsets for different regions affected by major floods. In accordance with the focus areas of inundations and flood impacts the CATI data are mainly located in the Elbe and Danube catchments in Germany. This suggests a regional subdivision of the empirical data set according to these
river basins for the investigation of spatial model transfer. In detail we partition the data set between the metropolitan area of Dresden (Saxony), the Elbe catchment (Saxony, Saxony-Anhalt, Thuringia), and the Danube catchment (Bavaria, Baden-Wuerttemberg), see Fig 1. This split is applied irrespective of the CATI survey campaign, and thus the regional sub-sets contain records from different flood events. The idea is to investigate examples with a small set of learning data for a small specific region (Dresden), a large learning data set from an extended region (Elbe catchment), and a small set of learning data from an
extended region (Danube catchment). The details for the learning and transfer applications are listed in Table 3. For these three regions we learn RF models using the selected variables and assess their predictive performance when transferred to the other regions. As we use a completely independent dataset for model transfer testing, no additional bootstrap on top of RF internal bootstrapping is required.





**Table 3.** Computational experiments for transfer applications

| Transfer experiment | Implementation | Learned on/applied to # buildings |
|---|---|---|
| d2E | Learned from Dresden and applied to Elbe | 310/1234 |
| d2D | Learned from Dresden and applied to Danube | 310/105 |
| E2D | Learned from Elbe and applied to Danube | 1234/105 |
| D2E | Learned from Danube and applied to Elbe | 105/1234 |

## 4   Results and Discussion

Random Forest OOB errors are sensitive to the choice of RF parameters *mtry* and *ntree*. From the variation of RF parameters we observe that OOB errors decrease with smaller values for *mtry* and larger numbers of trees in a forest (*ntree*), Fig. 3. The colored bands represent the 90-quantile range of OOB values from the 100 bootstrap repetitions for each RF algorithm

configuration and illustrate the inherent variability of input variables in the learning data set. The color code distinguishes the number of variables used to determine splits at each node (*mtry*). For *mtry* = 2 the smallest OOB errors are achieved throughout the variations in the number of trees (*ntree*). This value represents the lower bound of recommended values for *mtry* in RF regression models (Breiman, 2001). For smaller values of *mtry* less variables are considered for splitting the space of predictor variables, which reduces the correlation between individual trees of the forest. Further, increasing values of *ntree*

asymptotically approximate smaller OOB values. It appears that for the given data set OOB values are virtually stable above *ntree* = 7000. As the computational effort increases with larger forests it has to be balanced with improvements regarding predictive performance. Building on these results we use RF parameters *mtry* = 2 and *ntree* = 7000, which are comparable to those used by (Schröter et al., 2018).

### 4.1   Variable selection and predictive model learning

The numerical spatial measures (Table 2, and Appendix A1) evaluate properties of the building footprint geometries including area, perimeter, and elongation of main building axes. Accordingly some of these variables are strongly correlated (Fig. 4). The Spearman's rank correlation matrix of the variables confirms a high degree of correlation in the dataset, as for instance between Area, Perimeter and RadGyras. In contrast, the spatial measures are only slightly correlated with *wst* and *rloss*. The presence of multi-colinearity may influence the analysis of variable importance (Gregorutti et al., 2017). The robust importance analysis

uses different RF parameter settings and reports an average importance rank, which alleviates this problem.

The variable *wst* ranks first in the importance analysis (Fig. 5). This confirms common knowledge in flood loss modelling (Gerl et al., 2016; Smith, 1994). In comparison to *wst*, the numerical spatial measures of OSM building footprint geometries have clearly smaller importance values with relatively small differences between them. In terms of building characteristics, the rank order suggests that both spatial measures which express the size and extension of the building (e.g. Area, Perime-

ter) and spatial measures which describe building compactness and shape complexity (e.g. PARatio, RadGyras, LinSegInd, BoundRatio) add information to better estimate relative building loss.





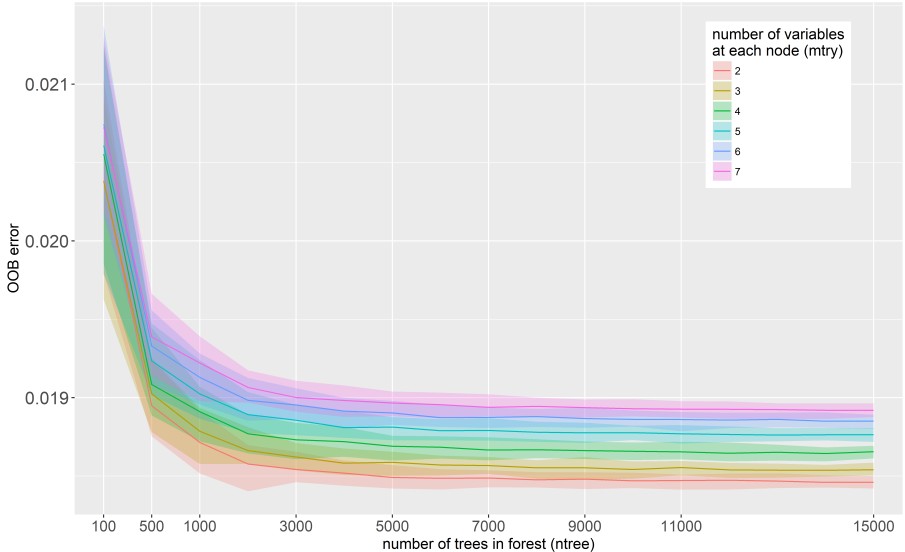

**Figure 3.** Out-of-Bag error for variations of mtry and ntree RF parameters. Color bands represent the variation range of OOB errors obtained from 100 bootstrap repetitions

The outcome of the variable importance analysis does not suggest a clear selection of features to be included in a predictive flood vulnerability model. The model predictive performance based assessment of variables uses an increasing number of variables following their ranking order of variable importance in the RF modelling. The predictive performance is quantified in terms of MAE, MBE, and MSE (Equations 1, 2, 3) for 100 bootstrap repetitions. While the MAE is decreasing when additional

variables are used with an overall minimum for a model using 6 variables, including more than 6 variables tends to increase MAE again (Fig. 6). However, regarding MBE these changes go in an opposite direction. We observe smallest MBE when only 2 variable are included. MBE then grows continuously for using up to 7 variables and then slightly reduces when more variables are used. The increase in precision expressed by the smaller MAE is accompanied with a reduction of accuracy reflected by an increasing MBE. This yields an almost balanced performance in terms of MSE for all models tested.

Looking into the sharpness of model predictions, the quantile range ($QR90$) is getting larger with an increasing number of model variables, which reflects larger uncertainty (Table 4. In terms of model reliability (HR), an increasing number of model variables achieves better performance statistics up to using 8 variables. The combination of both, $QR$ and $HR$, in the interval score ($IS$) shows a similar pattern.

On the basis of these assessments two model alternatives are selected for further analysis: Model A using 8 variables as

it provides the most reliable model predictions, and Model B using 6 variables which provide the highest precision and balance between accuracy and precision. In detail Model B uses the variables *wst*, PARatio, RadGyras, Area, LinSegInd, and BoundRatio. Model A, in addition, uses Perimeter and DegrComp as predictors.


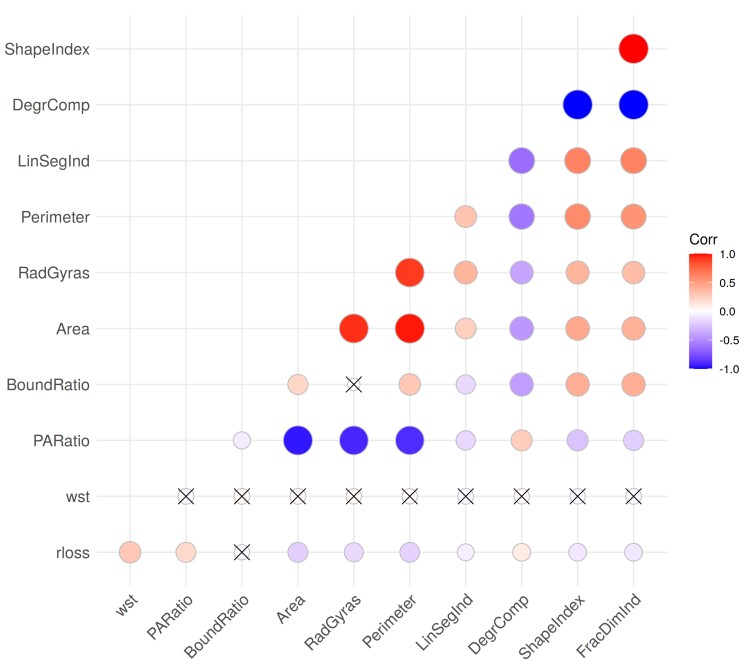

**Figure 4.** Spearman's correlation of model variables (significance level 1%), non significant correlations are crossed out.

**Table 4.** Model performance metrics for models using increasing number of variables including *wst*. Best performance values in bold.

| Model | MAE | MBE | MSE | QR | HR | IS |
|---|---|---|---|---|---|---|
| 2 variables | 0.0878 | **-0.0234** | 0.0230 | **0.2765** | 0.5864 | 7.9402 |
| 3 variables | 0.0853 | -0.0293 | 0.0226 | 0.2992 | 0.6301 | 7.1154 |
| 4 variables | 0.0843 | -0.0316 | 0.0224 | 0.3070 | 0.6433 | 6.8440 |
| 5 variables | 0.0840 | -0.0348 | 0.0227 | 0.3182 | 0.6533 | 6.7166 |
| 6 variables | **0.0826** | -0.0364 | **0.0222** | 0.3270 | 0.6622 | 6.5728 |
| 7 variables | 0.0830 | -0.0373 | 0.0225 | 0.3302 | 0.6614 | 6.5715 |
| 8 variables | 0.0839 | -0.0337 | 0.0224 | 0.3314 | **0.6640** | **6.3757** |
| 9 variables | 0.0841 | -0.0349 | 0.0226 | 0.3346 | 0.6639 | 6.3766 |
| 10 variables | 0.0844 | -0.0357 | 0.0228 | 0.3365 | 0.6631 | 6.4000 |
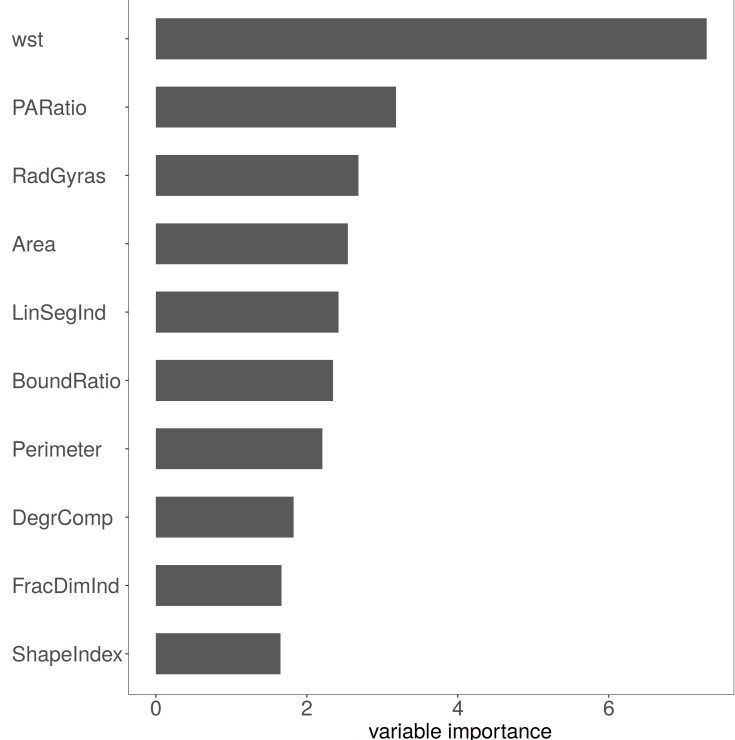

**Figure 5.** Average variable importance

## 4.2 Model predictive performance: model benchmarking

The OSM models A and B are benchmarked with a model that uses all information available from the CATI surveys as an upper benchmark (BMu) and a model that uses only water depth as predictor as a lower benchmark (BMl). The performance statistics achieved by models A and B for the complete data set (all events and regions) are slightly inferior to BMu but clearly

5   better than the outcomes of BMl (Fig. 7). Both models, A and B, give very similar performance statistics with slightly higher precision (smaller MAE) but larger bias (MBE) for model B. In contrast, model A provides more reliable predictions indicated by larger HR and smaller IS (Table 6). The randomized benchmark model (BMrm) achieves a better performance than BMl but is inferior to the models A and B (Fig. 7, Table 5). Hence, we are confident that the remaining uncertainty associated with the mapping of geolocations to building geometries is not affecting the outcomes of our analyses. Overall, we note that

10   including numerical spatial measures based on OSM building footprint geometries adds useful information to predict loss to residential buildings. The numerical spatial measures included in the models are all directly calculated using building footprint geometries. Therefore, a larger number of variables used for loss estimation does not imply increased efforts to collect data. From this perspective the cost of using model A or B is equal. The RF algorithm strives to reduce overfitting when large





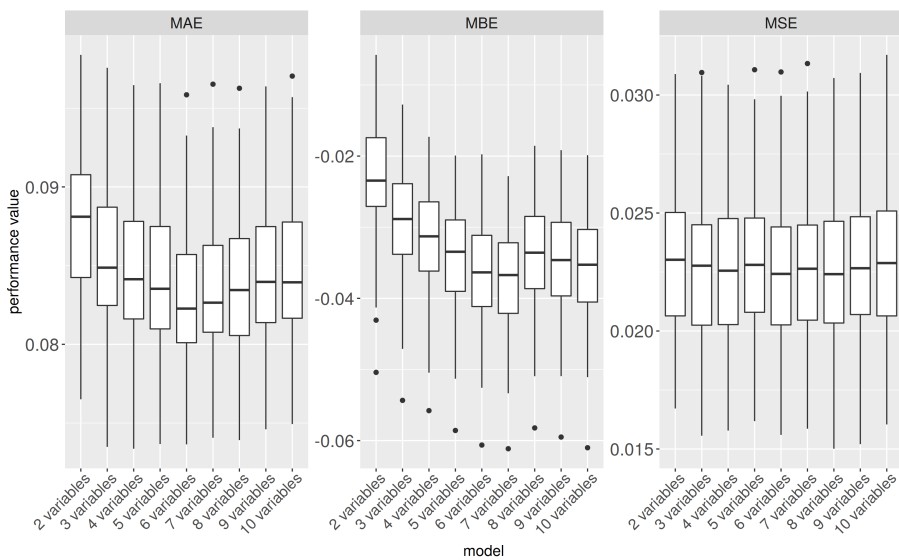

**Figure 6.** Predictive performance of models using an increasing number of variables. Smaller MAE and MSE values and MBE values close to 0 indicate better performance, c.f. equations 1 - 3.

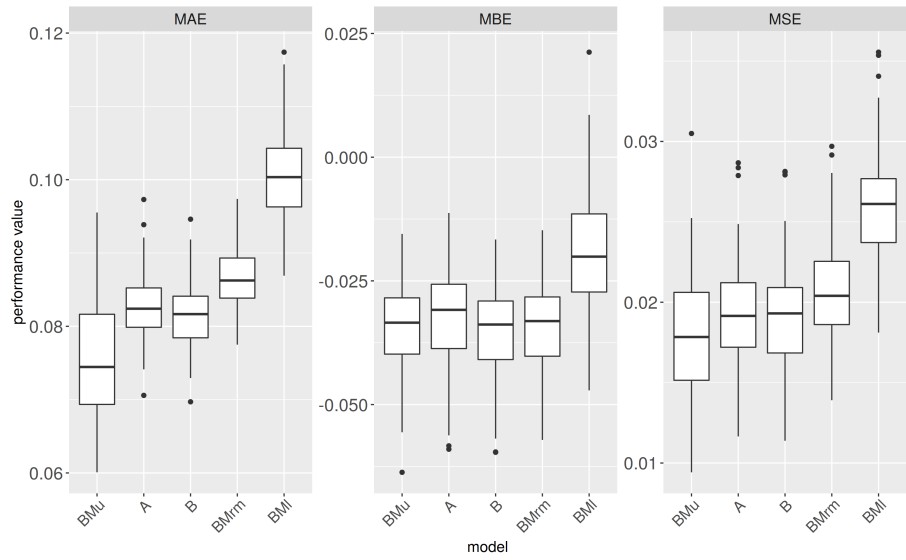

**Figure 7.** Performance metrics of OSM based models and benchmark models

numbers of predictors are included, and thus the parsimonious modelling principle can be relaxed. A possible negative effect of overfitting when using more predictors should manifest in spatial transfer applications.




**Table 5.** Model performance metrics

| Model | MAE | MBE | MSE | QR | HR | IS |
|---|---|---|---|---|---|---|
| BMu (Upper benchmark, all 23 predictors from CATI interviews) | 0.075 | -0.034 | 0.018 | 0.336 | 0.733 | 3.573 |
| A (7 numerical spatial measures derived from OSM plus water depth) | 0.083 | -0.032 | 0.019 | 0.322 | 0.699 | 6.022 |
| B (5 most important numerical spatial measures plus water depth) | 0.081 | -0.035 | 0.019 | 0.319 | 0.698 | 6.238 |
| BMrm (random match of CATI geolocation with OSM building polygons) | 0.087 | -0.034 | 0.021 | 0.319 | 0.688 | 6.535 |
| BMl (Lower benchmark, only water depth as predictor) | 0.100 | -0.019 | 0.026 | 0.177 | 0.490 | 10.107 |

## 4.3 Spatial transfer testing

The predictive performance of RF models is tested in regional transfer applications. For this purpose, the RF models A and B as well as the benchmark models BMu and BMl, as specified in the previous section, are learned using regional sub-sets of the data and applied to predict flood losses in a different region; see section 3.4 and Table 3 for details about the regional sub-division of data and spatial transfer experiments. Learning models with a regional sub-set of data and applying the models to other regions results in a drop of predictive performance in comparison to the case when the entire data-set is used for model learning, except for the case d2E (Fig. 8). In most of the learning/transfer cases, BMu scores best in terms of precision and reliability, represented by the performance metrics MAE, MSE, HR and IS. Using only *wst* as a predictor (BMl) produces less precise and less reliable predictions as indicated by larger MAE and MSE, as well as smaller HR and larger IS. While the performance of models A and B is very similar, model A, using 8 predictors, more reliably predicts residential loss (larger HR and smaller IS), and model B, using 6 predictors, provides more accurate (MBE closer to 0) and more precise predictions (smaller MAE and MSE). Hence, overfitting does not seem to be an issue when more input variables are used. In contrast to the model benchmark comparison (section 4.4) BMu and BMl do not entirely frame the RF model performance values. Instead, models A and B in some cases achieve better and in other cases worse performance statistics. Generally speaking, the predictive performance differs more strongly between the regional transfer settings than between the models (Fig. 8). This is more pronounced for precision and accuracy metrics (MAE, MBE and MSE) than for sharpness and reliability indicators (QR, HR and IS). Learning from the Dresden subset and transferring the model to the Elbe region (d2E) works best as is shown by the smallest MAE and MSE as well as a MBE closest to zero. Learning the models with the Danube sub-set and transferring them to the Elbe region (D2E) yields comparably small MAE and MSE values, but this is also the only case with a tendency to overestimate *rloss* resulting in a positive MBE. The models are struggling most to predict loss when they are learned with the Dresden sub-set and transferred to the Danube region (d2D) showing the lowest precision and accuracy. In turn, extending the learning subset to the Elbe region improves the transfer to the Danube (E2D). Concerning predictive uncertainty and reliability, learning with the Danube sub set yields large QRs, which however only partly cover the observed loss values reflected in comparably low HRs and high IS (D2E). Learning from Dresden/Elbe and transferring to Elbe or Danube (d2E, d2D, E2D) produces sharper predictions, but still the models differ in reliability, i.e. covering the observed values within their predictive uncertainty ranges (HR). In this respect, the upper benchmark model (BMu) performs best. The differences between models A


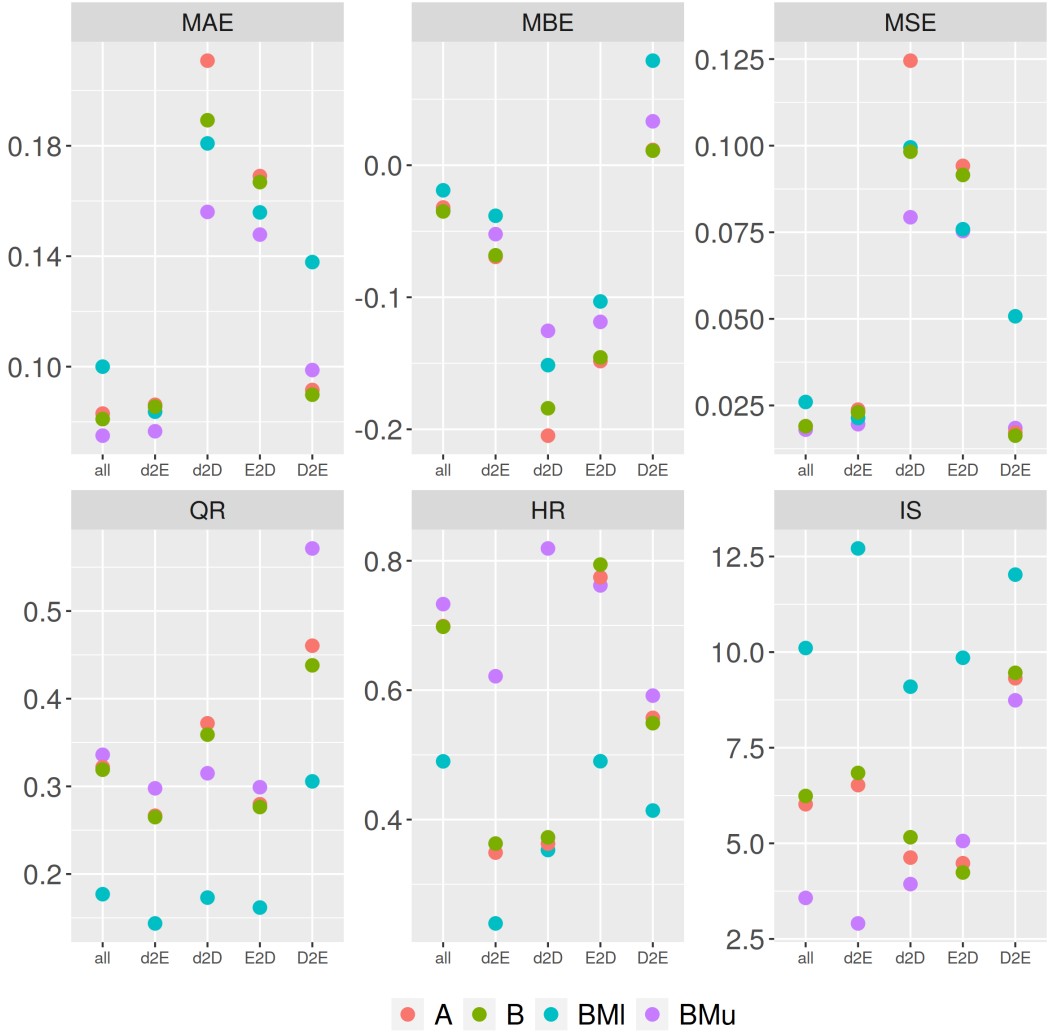

**Figure 8.** Model performance metrics in regional transfer. Models A and B based on spatial numerical measures calculated for OSM building footprint geometries, benchmark models BMl and BMu based on CATI survey data. Transfer experiments d2E, d2D, E2D, and D2E as described in Table 3.'all' refers to using all records from all regions, c.f. Table 5.

and B are small and both are better than the lower benchmark model (BMl) and almost similar to BMu for the transfer cases between the regions Elbe and Danube (E2D and D2E).

With 105 records the Danube data-set is the smallest sub-sample. It has a smaller variability and range of values for most numerical spatial measures in comparison to the Dresden and Elbe regional sets (Appendix A2). The geometric properties of the flood-affected residential buildings in the Danube region seem to differ from the affected residential buildings in the Elbe region. This can be attributed to different building practices in former East and West Germany as well as regional differences





in building types. With only 310 records, the Dresden sub-sample covers comparable ranges of observed variables as the Elbe sub-set (1234 records). Both sub-sets show largely similar relations between individual variables and *rloss*.

Still, the Danube sub-set includes relatively many records with high *rloss* values, which are distributed along the whole spectrum of above ground-level inundation depths (Appendix A2). In comparison, the Dresden sub-set comprises very few

cases with high relative loss which is partly related to differing inundation processes. In the Elbe and Danube catchments large areas have been flooded as a consequence of levee failures. Hence, the relationship of model variables to high *rloss* values cannot be learned from this sub-set, and thus is not represented well by the model. Therefore, this difference in the learning data may explain the positive bias introduced by learning the model in the Danube and transferring it to the Elbe, and, vice versa, the pronounced negative bias introduced by learning the model in Dresden and transferring it to the Danube region. Viewed from

a model performance perspective, the transfer applications show that a good agreement between learning and transfer data-sets (e.g. d2E) produces more precise and reliable predictions than the transfer to regions with pronounced differences (e.g. d2D, D2E). Still from the Danube region with limited ranges of variable values, it is possible to obtain relatively precise and accurate predictions of relative building loss. This suggests that a broad variability of observed *rloss* values in the learning data set is an important control for the predictive capability of the model in other regions. In contrast, small samples with limited variability

and only few records with high *rloss* values struggle with predicting *rloss* in other regions. This confirms insights that a model based on more heterogeneous data performs better when transferred in space (Wagenaar et al., 2018). Our findings also reveal that using numerical spatial measures derived from OSM building geometries does not resolve all problems of model transfer. But the spatial measures are useful proxy-variables for flood vulnerability characteristics of residential buildings. These proxies achieve comparable predictive performance as specific property level data sets as for instance collected via computer aided

telephone interview surveys. Using variables derived from OSM data increases the flexibility of the models to be applied in other regions because the accessibility and availability of OSM data reduces the effort of data collection, simplifies the preparation of input variables, and ensures consistency of input data. Achieving consistency of input data has been stressed to cause large efforts in model transfers (Jongman et al., 2012; Molinari et al., 2020). The suggested RF models are based on an ensemble approach, and thus provide a view to the predictive uncertainty of the model outputs. We have shown this to

be a valuable detail in assessing the reliability of model predictions in spatial transfers. In cases where model performance cannot be tested with local empirical evidence, using model ensembles has been shown to provide more skillful loss estimates (Figueiredo et al., 2018).

## 5  Conclusions

The transfer of flood vulnerability models to regions other than those for which they have been developed often comes with

reduced predictive performance. In this study we investigated the suitability of numerical spatial measures calculated for residential building footprint geometries, which are accessible from OpenStreetMap, to predict flood damage. Further we tested potential benefits from using widely available and consistent input data for the transfer of vulnerability models across regions. In this contribution we develop a new data-set based on open building data, which comprises of variables representing building



footprint geometric dimensions and shape complexity, and we devise novel flood vulnerability models for residential buildings. These models use only open data and can be applied to areas where information about the footprint geometry of residential buildings are available. The input variables of the models are easily extracted with an automatic process applicable to every type of building polygon. Hence, also other data sources, e.g. data derived from remote sensing, can be used besides the

OpenStreetMap data source which we have used in this study. The vulnerability models have been validated using empirical data of relative loss to residential buildings. Further, a benchmark comparison of the models has been conducted in spatial transfer applications. The geometric characteristics of building footprints provide useful proxies to describe building resistance to flood impacts and support flood loss estimation. OpenStreetMap is a highly popular and evolving data source with constantly increasing completeness and up to date data. In the future, the attributes of residential buildings are expected to provide

additional details which are of interest for the characterisation of building resistance to flooding. This includes for instance information about building type, roof type, number of floors, building material and opens further possibilities to refine the variables used for vulnerability modelling. These data could be further amended with other open data sources including socio-economic statistical data. In view of a large variability of flood loss on individual building level, vulnerability modelling for individual buildings remains challenging and is subject to large uncertainty. Advances to the understanding of damage

processes and the improvement of flood vulnerability modelling, hence requires an improved and extended monitoring of flood losses.

*Code and data availability.* Flood damage data of the 2005, 2006, 2010, 2011, and 2013 events along with instructions on how to access the data are available via the German flood damage database, HOWAS21 (http://howas21.gfz-potsdam.de/howas21/). Flood damage data of the 2002 event was partly funded by the reinsurance company Deutsche Rückversicherung (www.deutscherueck.de) and may be obtained

upon request. The surveys were supported by the German Research Network Natural Disasters (German Ministry of Education and Research (BMBF), 01SFR9969/5), the MEDIS project (BMBF; 0330688) the project "Hochwasser 2013" (BMBF; 13N13017), and by a joint venture between the German Research Centre for Geosciences GFZ, the University of Potsdam, and the Deutsche Ruckversicherung AG, Dusseldorf. OSM is an open data project and the cartographic information can be downloaded, altered and redistributed under the Open Data Commons Open Database License (ODbL) (contributors, 2020).

In the presented study, the geographic data were processed in PostgreSQL 12.2 with PostGIS 3.0.1 extension and R version 3.6.3 (2020-02-29) (R Core Team, 2020). The spatial measures were calculated in PostgreSQL and imported in to R for further processing. The RandomForest model was built and applied in R with the use of the following packages: randomForest 4.6-14 (Liaw and Wiener, 2002),sf 0.6-3 (Pebesma, 2018), reshape2_1.4.3 (Wickham, 2007), gdalUtilities_1.1.0 (O'Brien, 2020), rpostgis_1.4.3 (Bucklin and Basille, 2018), rgdal_1.4-8 (Bivand et al., 2019), raster_3.0-7 (Hijmans, 2019), RPostgreSQL_0.6-2 (Conway et al., 2017), tidyverse_1.3.0 (Wickham et al.,

30   2019).



# Appendix A

## A1 Definition and examples for numerical spatial measures

| | Variables | Equation | Meaning | Range and example values |
|---|---|---|---|---|
| 1 | Area (Area) | $Area$ | Area of the building in square meter | $[0\ m^2$       $\infty$ |
| 2 | Perimeter (Perimeter) | $Perimeter$ | Perimeter of the building in meter | $[0\ m$       $\infty$ |
| 3 | Degree of compactness (DegrComp) | $\dfrac{Area\ 4\pi}{Perimeter^2}$ | Compactness of the building shape, relative vicinity of the internal points, normalized to a circle | $[0$   = 0.10   = 0.70   = 1.00   $1]$ |
| 4 | Perimeter-area ratio (PARatio) | $Perimeter/Area$ | Simple measure of shape complexity, biased by building size | $[0$   = 0.20   = 0.30   $\infty$ |
| 5 | Shape index (ShapeIndex) | $\dfrac{Perimeter/4}{\sqrt{Area}}$ | Shape complexity, adjusted to building size, normalized to a square | $[1$   = 1.10   = 2.70   $\infty$ |
| 6 | Fractal dimension index (FracDimInd) | $\dfrac{2\ \ln(Perimeter/4)}{\ln Area}$ | Shape complexity, scaled from 1 to 2, adjusted to the size, norm to a square | $[1$   = 1.05   = 1.25   $2]$ |
| 7 | Radius of gyration (RadGyras) | $\sum_{i=1}^{n}\dfrac{dist(vertex_i - centroid)}{num\ of\ vertices}$ | Building extent and compactness in meter | $[0\ m$   = 8.20   = 31.05   = 63.25   $\infty$ |
| | | $Major, minor$ | Major and minor axes of the Minimum bounding rectangle | M   m |
| 8 | Linear segment indicator (LinSegInd) | $Major/minor$ | Elongation of the polygon, normalized to a square | $[1$   = 1.32   = 6.17   = 9.25   $\infty$ |
| 9 | Ratio of bounding rectangle area (BoundRatio) | $\dfrac{Bound\ rectangle\ Area}{Area}$ | Shape complexity, normalized to the hypothetical simplest polygon | $[1$   = 1.05   = 2.90   $\infty$ |



## A1 Scatterplots of numerical spatial measures and relative loss in regional sub-samples (Danube, Dresden, Elbe)

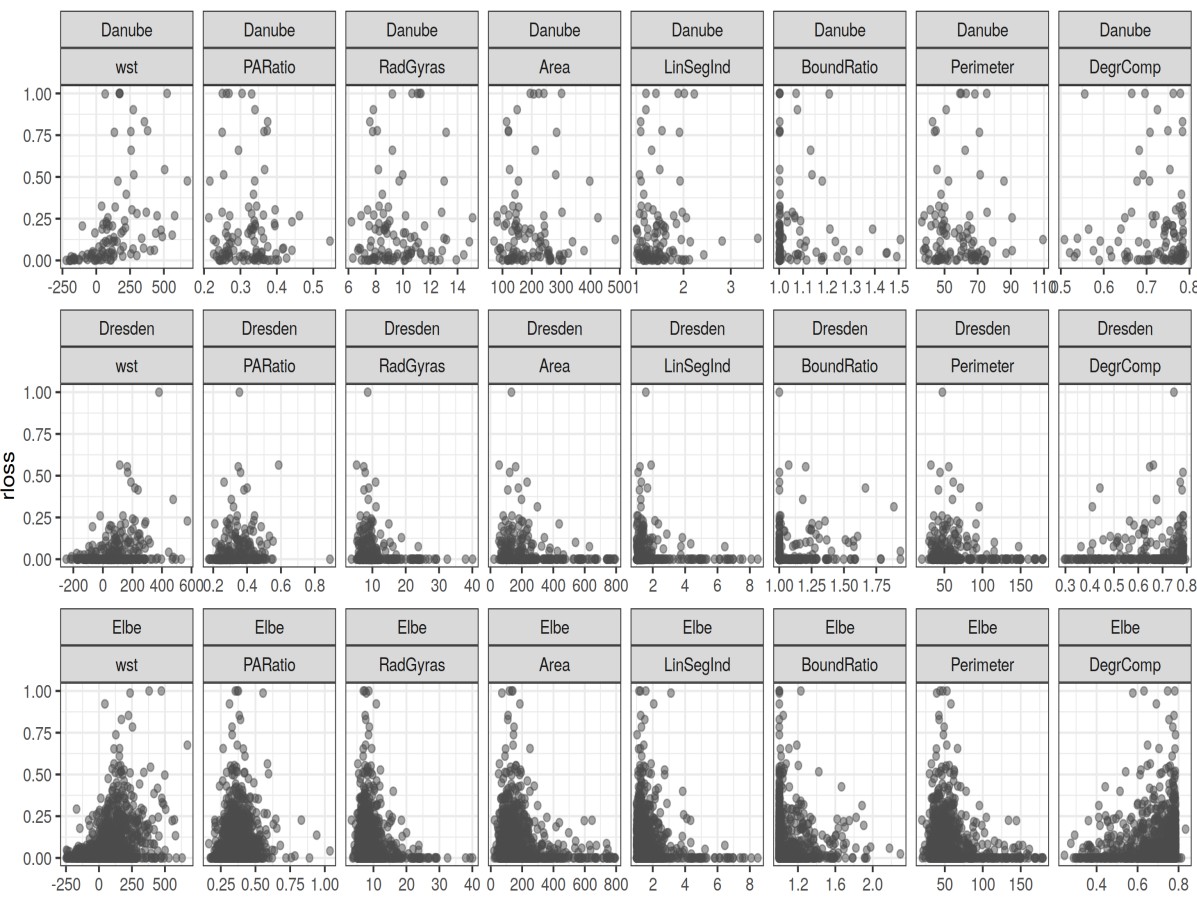

*Author contributions.* MC, MS, HK, KS

MC and KS conceived and designed the study. MC prepared and analysed the data with support from MS and KS. MC and KS wrote the first draft of the manuscript. HK helped guide the research through technical discussions. All authors reviewed the draft manuscript and contributed to the final version.

*Competing interests.* The authors declare that no competing interests are present



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
