# Peer review of "Are OpenStreetMap building data useful for flood vulnerability modelling?"

_Natural Hazards and Earth System Sciences, 2020_

## Referee Comment (RC1) · Anonymous Referee #1 · 20 Jul 2020

In this paper the Authors analyzed the possible contribution of using Open Street Map (OSM) data for enhancing the predictive performance and transferability in space of multi-variable flood damage models for the residential sector. To this purpose, they built a dataset by combining empirical observations from historical flood events in Germany and data derived from OSM, with the latter essentially related to building footprint geometry. Random forest regression models (RFM) were then learned on this dataset using regional sub-sets and were tested for predicting flood losses in other regions. The manuscript is overall well written and presented and the topic perfectly fits the scope of NHESS, following the path of similar papers published in the Journal in recent years. However, in my opinion, the study suffers from a main methodological criticality, i.e. the representativity of the new additional parameters in correctly characterizing

the building vulnerability to floods. Indeed, the nine selected parameters derived from OSM used for learning RFM were only related to the shape and extension of the building footprint area (with an obvious high correlation among them), neglecting instead other fundamental vulnerability variables, e.g. building material and type, presence of a basement, etc. As it is well known and understandable, footprint geometry has a high influence in determining flood losses; however, as shown in previous studies, the observed damage variability depends on many (hazard and) vulnerability factors, which should not be neglected for a comprehensive modelling of flood damages. This becomes even more important when we consider the problem of the spatial transferability of empirical damage models. For instance, we may have two regions which have similar characteristics in terms of footprint geometry, but very different construction types: in this case, an OSM-based multi-variable model would be totally unreliable. For this reason, the main question that the Authors asked in the title "Are new open building data useful for flood vulnerability modelling?" is a bit pretentious, given that the answer is quite obvious if they limit their analysis on including only the nine additional variables listed in Table 2. For the same reason, also the results shown in Section 4 are expected; moreover, these indicated that the consideration of all the new footprint parameters does not actually greatly improve model performances (Table 4). Also the variable importance shown in Figure 5 is only partly informative: it basically says that water depth is more important than building shape and extension, but this is already known (and also shown in similar studies, e.g. Wagenaar et al. 2017, Amadio et al. 2019, both published in NHESS). The Authors are right in saying that information on building attributes in the OSM database are scarce and not useful for the kind of analysis they performed in their study. However, they could have exploited other public databases existing in Germany (e.g. cadastral, city planning maps, etc.) for building a more complete dataset. Therefore, I would suggest to the Authors to consider this possibility and repeat the same analysis in order to have more interesting results for improving our knowledge on flood damage modelling.

Specific comments: - P1.L13-15 and L16-18: based on previous general comments,

I find these sentences potentially dangerous. - P1.L17: what do you mean with "consistent"? - P3.L14-16 and L29-30: you said that one of the main aims of the paper is to understand which building variables are useful to characterize building vulnerability, but you actually investigated only footprint-related indicators, which only capture part of the overall building vulnerability. - P3.L32: typo "modelsi". - Figure 2. Acronyms shown in the figure are defined in the text of the paper, but it would be better to report them also in the figure caption. - P10.L19: "this analyses" -> "these analyses". - P10.L31: missing parenthesis after "Table 2". - P12.L4: please rewrite this sentence. - P12.L8: remove comma after "reasoning is" - P12.L5-14: this part should be moved to the previous section. - P12.L18-19: please rewrite this sentence. - P13.L21-26: as discussed in general comments, this result is expected and only partly informative, because you neglected other important vulnerability variables. - P14.L1-2: this is also expected and due to the selected variables. - P14.L11: missing parenthesis after "Table 4". - P15.L15: you finally chose the models with 6 and 8 variables (as the best performing ones). This is fine, but, actually, the variability in the performance indicators is very small (this is also due to the used variables), and probably you could have opt for the simpler models. - Figure A2 should be moved to the main text (and not in the Appendix) and discussed in more detail for the interpretation of the results. - P19.L6: I think this point deserves more discussion and analysis (see also my general comments). You just mention it. - P20.L17-22: as in the abstract, these are potentially dangerous statements. - P21.L7-8: as in the abstract, these are potentially dangerous statements. - P12. L9-10: I agree and this is what I suggest you to do (you can use information from other public databases to be merged with data coming from OSM). Otherwise, at present, this study provides partial (and potentially misleading) insights for flood damage modelling.
* * *

---

## Referee Comment (RC2) · Anonymous Referee #2 · 27 Aug 2020

Dear Editor / Authors,

Thank you very much for the opportunity to act as a review on your very interesting paper which I believe is very much worthy of publication. The paper is really addressing two questions:

1. Can a Machine Learning Technique provide good predictions / estimates of flood damage using Open Data in the same location? 2. Can the results of the open data model then be transferred to another location, and how good are the predictions / estimates?

These are both very interesting questions and the paper addresses it well.

I am not an expert on Machine Learning Algorithms such as Random Forests, so I

found myself having to do some background reading to understand the methodology. I suspect this may be a problem for general readers such as myself. If I look at Figure 2, this presents the steps well, but I wonder if there is a possibility for a clearer explanation in a few lines of the purpose of the Random Forest approach. Almost along the lines of "Random Forests are able to make predictions of flood loss by creating numerous decision trees, based on the random selection of decision nodes"?

The methodology and analysis is well described and the figures are clear and well labelled.

The only reservation is that the conclusion and abstract could be strengthened because I think it's an interesting paper. You write in the abstract that "However, our results show that using numerical spatial measures derived from OpenStreetMap building geometries does not resolve all problems of model transfer." You say the models are useful, but I don't get a sense from the abstract or conclusion that you are very confident in this. Similarly, if I were to jump to the conclusions, I don't get a clear sense of how well the open data models work, first of all, in the same location, and when you transfer them to different location, without having to go back into the results and discussion. I feel the conclusion should be clearer here to state what was the real value in using OpenStreetMap data.

I have some grammar / typo suggestions.

Page 2 Line 11 - "Modeling". In the rest of the paper, you use modelling – please be consistent with the spelling, except in the references where titles are quoted directly. Line 16 – advance, not advancement Line 32 – "Tree-based". I would use a hyphen here Page 3 Line 11 "It was shown that particularly geometric information about buildings as for instance building area and height are useful variables to describe building characteristics relevant for estimating flood losses (Schröter et al., 2018)." I think this sentence could be simplified – "It was shown that geometric information such as building area and height are useful . . ." Line 14 – "building footprint geometry" – the word

footprint or geometry can be removed Line 23 – "most of civil and common uses" – of can be deleted. Line 33 – modelsi – please correct the typo Page 7 Figure 1 - could you present the locations all on one map of Germany? I appreciate this would mean overlaying Dresden and the Elbe, but three maps seems unnecessary. Line 10 – The spatial measures are described in a table – I think the paragraph can be eliminated as the table repeats the information. Page 9 – Please correct transver in the figure. Page 20 Line 30 - Please correct OpenStreeMap.

---

## Author Comment (AC1) · 18 Sep 2020

**Response to Referee 1**

We would like to thank the referee for the time and effort put into reviewing the manuscript. This response (R) carefully addresses all the comments (C). Where applicable, changes are proposed to the manuscript accordingly.

*C: In this paper the Authors analyzed the possible contribution of using Open Street Map (OSM) data for enhancing the predictive performance and transferability in space of multi-variable flood damage models for the residential sector. To this purpose, they built a data-set by combining empirical observations from historical flood events in Germany and data derived from OSM, with the latter essentially related to building footprint geometry. Random forest regression models (RFM) were then learned on this data-set using regional sub-sets and were tested for predicting flood losses in other regions. The manuscript is overall well written and presented and the topic perfectly fits the scope of NHESS, following the path of similar papers published in the Journal in recent years.*

R: We thank the reviewer for this basically positive evaluation.

*C: However, in my opinion, the study suffers from a main methodological criticality, i.e. the representativity of the new additional parameters in correctly characterizing the building vulnerability to floods. Indeed, the nine selected parameters derived from OSM used for learning RFM were only related to the shape and extension of the building footprint area (with an obvious high correlation among them), neglecting instead other fundamental vulnerability variables, e.g. building material and type, presence of a basement, etc. As it is well known and understandable, footprint geometry has a highinfluence in determining flood losses; however, as shown in previous studies, the observed damage variability depends on many (hazard and) vulnerability factors, which should not be neglected for a comprehensive modelling of flood damages.*

R: We agree with the reviewer that building vulnerability is determined by diverse influencing factors and acknowledge that understanding how building vulnerability can be correctly characterized implies valid and relevant research questions.
However, the generic research question of our study is on assessing how new promising data sources like volunteered geographic information and open data can help to tackle challenges in natural hazard research. Specifically, we focus on OpenStreetMap as a potential data source for flood vulnerability modeling in its current state. We focus on OpenStreetMap, since it is the most comprehensive open data containing building footprints data of good quality. With this in mind, we do not aim to characterize building vulnerability as comprehensively as possible, but rather to see what is possible in terms of building flood vulnerability modeling with the available OpenStreetMap data. This knowledge will support future studies on building flood vulnerability which may investigate additional building characteristics and also the appropriateness of other data sources.
To make this dedicated focus clearer we suggest to change the title into: 'Are OpenStreetMap building data useful for flood vulnerability modelling?'. In addition we will make changes to the abstract, the introduction, discussion and conclusions as detailed in the following responses..

*C: This becomes even more important when we consider the problem of the spatial transferability of empirical damage models. For instance, we may have two regions which have sim ilar characteristics in terms of footprint geometry, but very different construction types: in this case, an OSM-based multi-variable model would be totally unreliable. For this reason, the main question that the Authors asked*

*in the title "Are new open building data useful for flood vulnerability modelling?" is a bit pretentious, given that the answer is quite obvious if they limit their analysis on including only the nine additional variables listed in Table 2.*

R: As said in the answer to the previous comment we will adjust the title to make the focus on research objective of this study clearer. The suggested title also fits better to our research hypothesis p3L26-27. In addition we will state research objective i) more precisely: 'understand which building geometry variables are useful to describe building vulnerability'. (p3L29)

*C: For the same reason, also the results shown in Section 4 are expected; moreover, these indicated that the consideration of all the new footprint parameters does not actually greatly improve model performances (Table 4). Also the variable importance shown in Figure 5 is only partly informative: it basically says that water depth is more important than building shape and extension, but this is already known (and also shown in similar studies, e.g. Wagenaar et al. 2017, Amadio et al. 2019, both published in NHESS).*

R: Figure 5 represents the outcomes of an intermediate step of our data analyses workflow. The purpose for the assessment of variable importance is to get a basic understanding of the suitability of individual predictors in a highly correlated data set. In this regard the assessment of variable importance adds to the correlation analyses, but (we agree with reviewer) it does not reveal fundamentally new findings. We suggest to remove this figure from the manuscript and refer to the to the results of the assessment of variable importance in the text P13.LXX

*C: The Authors are right in saying that information on building attributes in the OSM database are scarce and not useful for the kind of analysis they performed in their study. However, they could have exploited other public databases existing in Germany (e.g. cadastral, city planning maps, etc.) for building a more complete data-set. Therefore, I would suggest to the Authors to consider this possibility and repeat the same analysis in order to have more interesting results for improving our knowledge on flood damage modelling.*

R: We fully agree that this would be an interesting research but it is beyond the scope of this study. We mention this perspective in our conclusions P21L9-10.

*Specific comments:*

*C: P1.L13-15 and L16-18: based on previous general comments, I find these sentences potentially dangerous.*

R: We will add further details to the abstract to better frame these statements to the context of this study and emphasize requirements for spatial model transfer. We suggest to rephrase as follows:
This regional split-sample validation approach reveals that the predictive performance of models based on OpenStreetMap building geometry data is comparable to alternative multi-variable models, which use comprehensive and detailed information about preparedness, socio-economic status and other aspects of residential building vulnerability. Still, the transfer of these models to other regions should include a test of model performance using independent local flood loss data.

*C: P1.L17: what do you mean with "consistent"?*

R: We use the word consistent with the meaning that something is accordant or compatible, i.e. adhering to the same definitions. With respect to OSM data this implies that the model variables and underlying data are based on the same data model, have the same definition, format, unit, etc.

*C: P3.L14-16 and L29-30: you said that one of the main aims of the paper is to understand which building variables are useful to characterize building vulnerability, but you actually investigated only footprint-related indicators, which only capture part of the overall building vulnerability.*

R: As said in the above responses, the focus of this study is on the use of OpenStreetMap data in its current status for flood vulnerability modeling. To make this dedicated focus clearer we suggest to change the title into: 'Are OpenStreetMap building data useful for flood vulnerability modelling?', and will state research objective i) more precisely: 'understand which building geometry variables are useful to describe building vulnerability'. (p3L29)

*C: P3.L32: typo "modelsi".*

R: will be corrected

*C: Figure 2. Acronyms shown in the figure are defined in the text of the paper, but it would be better to report them also in the figure caption.*

R: We will adjust the figure caption to include the abbreviations:
'Fig. 2: Data pre-processing, model learning and model transfer workflow, with BMu (upper benchmark model), BMl (lower benchmark model), BMrm (Benchmark model with random match of interview locations with OSM building data), A (Random Forest model using 8 predictors), B ( Random Forest model using 8 predictors), and model transfers d2E (learning with Dresden and predictions for Elbe), d2D (learning with Dresden and predictions for Danube), E2D (learning with Elbe and predictions for Danube), D2E (learning with Danube and predictions for Elbe)'

*C: P10.L19: "this analyses" -> "these analyses".*

*R:will be corrected*

*C: P10.L31: missing parenthesis after "Table 2".*

R: will be corrected

*C: P12.L4: please rewrite this sentence.*

R: we will rephrase the sentence "Further, an independent assessment of OSM based vulnerability model performance we consider two benchmark models.".
into:
"Further, for an independent assessment of OSM based vulnerability model performance we consider two benchmark models.".

*C: P12.L8: remove comma after "reasoning is"*

R: will be corrected

*C: P12.L5-14: this part should be moved to the previous section.*

R: We agree with the reviewer and will move this paragraph to the previous section (3.2 Predictive model learning)

*C: P12.L18-19: please rewrite this sentence.*

R:We suggest to rewrite this sentence as follows:
The CATI data are mainly located in the Elbe and Danube catchments in Germany, which are the regions mostly affected by inundations and flood impacts.

*C: P13.L21-26: as discussed in general comments, this result is expected and only partly informative, because you neglected other important vulnerability variables.*

R: As stated above this is beyond the scope of our study. We will add insights from other recent studies (e.g. Wagnenaar 2017, Vogel et al. 2018, Carisi et al. 2018, Amadio et al. 2019) about the usefulness of other potential predictors for building vulnerability to the discussion.

*C: P14.L1-2: this is also expected and due to the selected variables.*

R: The assessment of variable importance using Random Forests has been included to the data analyses workflow because, in addition to the correlation analysis, it accounts for variable interaction effects. We report this outcome for the sake of completeness and transparency. As stated in our above response we agree with reviewer that it does not reveal fundamentally new findings and suggest to remove this figure from the manuscript.

*C: P14.L11: missing parenthesis after "Table 4".*

R: will be corrected

*C: P15.L15: you finally chose the models with 6 and 8 variables (as the bestperforming ones). This is fine, but, actually, the variability in the performance indicators is very small (this is also due to the used variables), and probably you could have opt for the simpler models.*

R: We agree that the differences in performance between the models are not pronounced. The calculation of the variables from building footprints is done automatically and does not require additional effort for data retrieval and formatting. Therefore, we base our selection of candidate models on objective measures of model performance.

*C: Figure A2 should be moved to the main text (and not in the Appendix) and discussed in more detail for the interpretation of the results.*

R: We will follow the suggestion of the reviewer and include Figure A2 as a new Figure 9 to the manuscript. We will expand the discussion about regional differences visible for the regional sub-samples in the text.

*C: P19.L6: I think this point deserves more discussion and analysis (see also my general comments). You just mention it.*

R: As stated in the previous answer, we will expand the discussion about difference in regional sub-samples.

*C: P20.L17-22: as in the abstract, these are potentially dangerous statements.*

R: As stated above we will adjust the title to make the focus on research objective of this study clearer and refine our research objective i): 'understand which building geometry variables are useful to describe building vulnerability'. (p3L29).
At this point we will stress the idea of spatial measures as proxy variables more clearly.:
As not many variables of building characteristics are available from OSM data, the spatial measures calculated from building footprint serve as a sort of proxy variables for these unavailable details.

*C: P21.L7-8: as in the abstract, these are potentially dangerous statements.*

R: In line with the previous answer we will also rephrase this sentence to emphasize the idea of spatial measures as proxy variables for unavailable details about building vulnerability characteristics:
The geometric characteristics of building footprints serve as proxy variables for building resistance to flood impacts and are of use for flood loss estimation.

*C: P12. L9-10: I agree and this is what I suggest you to do (you can use information from other public databases to be merged with data coming from OSM). Otherwise, at present, this study provides partial (and potentially misleading) insights for flood damage modelling.*

R: Indeed this is another interesting research study. We think, that with the redefined title and more precise formulation of our research objectives this type of analyses is out of scope of this study.

**References**
Amadio, M., Scorzini, A. R., Carisi, F., Essenfelder, A. H., Domeneghetti, A., Mysiak, J. and Castellarin, A.: Testing empirical and synthetic flood damage models: the case of Italy, Natural Hazards and Earth System Sciences, 19(3), 661–678, doi:https://doi.org/10.5194/nhess-19-661-2019, 2019.
Carisi, F., Schröter, K., Domeneghetti, A., Kreibich, H. and Castellarin, A.: Development and assessment of uni- and multivariable flood loss models for Emilia-Romagna (Italy), Natural Hazards and Earth System Sciences, 18(7), 2057–2079, doi:https://doi.org/10.5194/nhess-18-2057-2018, 2018.
Vogel, K., Weise, L., Schröter, K. and Thieken, A. H.: Identifying Driving Factors in Flood-Damaging Processes Using Graphical Models, Water Resources Research, 54(11), 8864–8889, doi:10.1029/2018WR022858, 2018.
Wagenaar, D., de Jong, J. and Bouwer, L. M.: Multi-variable flood damage modelling with limited data using supervised learning approaches, Nat. Hazards Earth Syst. Sci., 17(9), 1683–1696, doi:10.5194/nhess-17-1683-2017, 2017.

---

## Author Comment (AC2) · 18 Sep 2020

**Response to Referee 2**

We would like to thank the referee for the time and effort put into reviewing the manuscript. This response carefully addresses all the comments. Where applicable, changes are proposed to the manuscript accordingly.

*C: Dear Editor / Authors,*
*Thank you very much for the opportunity to act as a review on your very interesting paper which I believe is very much worthy of publication. The paper is really addressing two questions: 1. Can a Machine Learning Technique provide good predictions / estimates of flood damage using Open Data in the same location? 2. Can the results of the open data model then be transferred to another location, and how good are the predictions /estimates? These are both very interesting questions and the paper addresses it well.*

R: We thank the reviewer for his positive comments.

*C: I am not an expert on Machine Learning Algorithms such as Random Forests, so I found myself having to do some background reading to understand the methodology. I suspect this may be a problem for general readers such as myself. If I look at Figure 2, this presents the steps well, but I wonder if there is a possibility for a clearer explanation in a few lines of the purpose of the Random Forest approach.*

R: We will add an explanation of the Random Forest approach

*C: Almost along the lines of "Random Forests are able to make predictions of flood loss by creating numerous decision trees, based on the random selection of decision nodes"?*

R: We will add the following sentence to the paragraph on RF algorithm in section 3:
'RF make predictions based on a large number of decision trees, i.e. a forest,
which is learned by randomly selecting the variables considered for the splitting of the feature space of the data.'

*C: The methodology and analysis is well described and the figures are clear and well labelled.*

R: Thanks again for the positive evaluation.

*C: The only reservation is that the conclusion and abstract could be strengthened because I think it's an interesting paper. You write in the abstract that "However, our results show that using numerical spatial measures derived from OpenStreetMap building geometries does not resolve all problems of model transfer." You say the models are useful, but I don't get a sense from the abstract or conclusion that you are very confident in this. Similarly, if I were to jump to the conclusions, I don't get a clear sense of how well the open data models work, first of all, in the same location, and when you transfer them to different location, without having to go back into the results and discussion. I feel the conclusion should be clearer here to state what was the real value in using OpenStreetMap data.*

R: We will rework the abstract and the conclusion section to be more self contained about the key outcomes of the research. We will include comparative statements about model performance of the OpenStreetMap based models and the benchmark models, e.g:

"Including numerical spatial measures based on OpenStreetMap building footprint geometries reduces model prediction errors (MAE by 20% and MSE by 25%) as well as increases the reliability of model predictions by a factor of 1.4 in terms of the Hit Rate when compared to a model that uses only inundation depth. This also applies to model transfer applications."

*I have some grammar / typo suggestions.*
*C: Page 2 Line 11 - "Modeling". In the rest of the paper, you use modelling – please be consistent with the spelling, except in the references where titles are quoted directly.*

R: will be corrected

*C: Line 16 – advance, not advancement*

R: will be corrected

*C: Line 32 – "Tree-based". I would use a hyphen here*

R: will be corrected

*C: Page 3 Line 11 "It was shown that particularly geometric information about buildings as for instance building area and height are useful variables to describe building characteristics relevant for estimating flood losses (Schröter et al., 2018)." I think this sentence could be simplified – "It was shown that geometric information such as building area and height are useful . . ."*

R: Thank you for the suggestion. We will rephrase the sentence accordingly.

*C: Line 14 – "building footprint geometry" – the word footprint or geometry can be removed*

R: we will remove the word 'geometry', also in other occurrences of the manuscript.

C: Line 23 – "most of civil and common uses" – of can be deleted.

R: we will delete this part.

*C: Line 33 – modelsi – please correct the typo*

R: will be corrected

*C: Page 7 Figure 1 – could you present the locations all on one map of Germany? I appreciate this would mean overlaying Dresden and the Elbe, but three maps seems unnecessary.*

R: We will rework the figure accordingly.

*C: Line 10 – The spatial measures are described in a table – I think the paragraph can be eliminated as the table repeats the information.*

R: We will shorten the paragraph and remove redundant information with Table 2.

*C: Page 9 – Please correct transver in the figure. Page 20*

*R: will be corrected*

*C: Line 30 - Please correct OpenStreeMap.*

*R: will be corrected*

---

## Referee Report (RR1)

Journal: NHESS
Title: **Are OpenStreetMap building data useful for flood vulnerability modelling?**
Author(s): Cerri et al.
MS No.: NHESS-2020-206
MS Type: Research Article
**Iteration: Second review**

The objective of the paper is to evaluate whether numerical spatial measures derived from OSM building footprints provide useful information for the estimation of flood losses to residential buildings, and to understand whether the use of such data can improve the spatial transferability of flood damage models. Specifically, three research objectives are identified by the authors: i) to understand which building geometry related variables are useful to describe building vulnerability, ii) to learn predictive flood vulnerability models, and iii) to test and evaluate model transfer across regions.
To do this, they created a new data-set by combining empirical data from historical flood events in Germany and data derived from OSM. Random forest regression models (RFM) were then
learnt on OSM data, and compared with similar models learnt on a more comprehensive set of damage explicative variables, rather than by considering the water depth as the only explicative variable. The comparison was done for the whole set of data, and by using regional sub-sets for predicting flood losses in other regions.

The manuscript is overall well written and presented. Figures and tables are clear, and conclusions are drawn from results. I do not have specific comments. But, some general considerations that could improve the quality of the paper.

1) According to my understanding, authors reach a very important conclusion. In data-scarce regions, where no "local" information is available on building vulnerability, the use of OSM derived spatial measures to learn multi-variable models gives comparable performance to alternative multi-variables models (which use comprehensive and detailed information about preparedness, socio-economic status and other aspects of building vulnerability), and better performance than models based only on water depth. Second, although the use of OSM does not resolve at all the problem of transferability (i.e. models remain strongly context specific and can be transferred only to regions with similar building geometric features than the calibration one), it supports transferability, by guarantee consistency in input variables between the implementation and the derivation context. Such considerations are explicitly written in the abstract, but they are implicit/hidden in the text; on the contrary, they should be highlighted to improve the usability of paper results.

2) Given the high correlation among OSM derived spatial measures and the impossibility to rank the weight of such variables in shaping damage, the fact that OSM derived models perform similar than other MV models could be due, on the one hand, on the fact that spatial measures serve as proxy variables for other vulnerability parameters not considered by the model (as suggested by authors), but also on the fact that damage mostly depends on extensive variables, like area and perimeter, while the role of other variables (both geometric or socio-economic/technical) is negligible. From this perspective, it could be useful to evaluate also the performance of a model based only on three/two variables: water depth, area (and perimeter).

Minor comments:
- Could model uncertainty also be linked to differences in building geometries derived from present OSM data and building geometries at the time of the event?
- Table 2 and Table A1 do not help in understanding the meaning of the different spatial mesures. Meaning of extreme values should be included in Table 2, at least.

---

## Author Response (AR2)

**Response to Referee 4**

We would like to thank the referee for the time and effort put into reviewing the manuscript. This response (R) carefully addresses the comments (C). Where applicable, changes are proposed to the manuscript accordingly.

*C: According to my understanding, authors reach a very important conclusion. In data-scarce regions, where no "local" information is available on building vulnerability, the use of OSM derived spatial measures to learn multi-variable models gives comparable performance to alternative multi-variables models (which use comprehensive and detailed information about preparedness, socio-economic status and other aspects of building vulnerability), and better performance than models based only on water depth. Second, although the use of OSM does not resolve at all the problem of transferability (i.e. models remain strongly context specific and can be transferred only to regions with similar building geometric features than the calibration one), it supports transferability, by guarantee consistency in input variables between the implementation and the derivation context. Such considerations are explicitly written in the abstract, but they are implicit/hidden in the text; on the contrary, they should be highlighted to improve the usability of paper results.*

R: We agree with the reviewer, and we will emphasise these outcomes in the conclusions as suggested in the track changes manuscript.

*C: Given the high correlation among OSM derived spatial measures and the impossibility to rank the weight of such variables in shaping damage, the fact that OSM derived models perform similar than other MV models could be due, on the one hand, on the fact that spatial measures serve as proxy variables for other vulnerability parameters not considered by the model (as suggested by authors), but also on the fact that damage mostly depends on extensive variables, like area and perimeter, while the role of other variables (both geometric or socio-economic/technical) is negligible. From this perspective, it could be useful to evaluate also the performance of a model based only on three/two variables: water depth, area (and perimeter)*

R: We support this suggestion. The performance of models using different numbers of input variables derived from building geometries is anlaysed in section 4.1 of the paper and the results are reported in Figure 5 and Table 4, e.g. the model using 2 variables is based on water depth and Perimeter-area ratio. For clarification, we suggest to include these details to the text of section 4.1 and also give these details in the caption of Table 4.

*C: Could model uncertainty also be linked to differences in building geometries derived from present OSM data and building geometries at the time of the event?*

R: This is an interesting point. OSM Data have been retrieved in 2017 and the surveys of affected households have been conducted in the period from 2002 to 2014. In theory, some of the buildings may have changed in terms of geometry by for instance retrofitting, demolition and new construction. However, we consider that this only applies to a minor number of buildings. Instead, other sources of uncertainty play a more important role. For instance, the uncertainty related to limited detail of geo-location information which affects the matching of telephone survey data to OSM building objects. This has been addressed with the benchmarking model Brm.

*C: Table 2 and Table A1 do not help in understanding the meaning of the different spatial mesures. Meaning of extreme values should be included in Table 2, at least*

R: We think the key information to understand the meaning of the different spatial measures is given in Table 2 and the table with geometry examples in Annex A1. However, some details may be hidden in the text of Table 2 and not directly apparent to the reader. As we have been asked in the previous round of reviews to remove the description of spatial measures from the text, we suggest changing the structure of Table 2. We propose to include an additional column, which gives information about the ranges of the different variables.